# Structure As Search: Unsupervised Permutation Learning for Combinatorial Optimization

## Abstract

We propose a non-autoregressive framework for the Travelling Salesman Problem where solutions emerge directly from learned permutations, without requiring explicit search. By applying a similarity transformation to Hamiltonian cycles, the model learns to approximate permutation matrices via continuous relaxations. Our unsupervised approach achieves competitive performance against classical heuristics, demonstrating that the inherent structure of the problem can effectively guide combinatorial optimization without sequential decision-making.

## 1 Introduction

How can we solve the Travelling Salesman Problem (TSP), a classic NP-hard problem in combinatorial optimization? Given a set of cities and pairwise distances, the goal is to find the shortest tour that visits each city exactly once and returns to the starting point. A widely held view is that to obtain reasonably good solutions, some form of search is almost always necessary (Applegate, 2006). Whether through greedy construction, local improvement, or other heuristics, search remains the standard approach. Exact solvers such as Concorde can solve moderate-sized instances optimally, but their exponential runtime limits scalability (Applegate et al., 2003). To handle larger instances, local search techniques such as Lin–Kernighan–Helsgaun (LKH) (Lin & Kernighan, 1973; Helsgaun, 2000) have long been used as practical heuristics.

While classical methods remain highly effective, they rely on meticulously designed rules and hand-crafted techniques. In contrast, neural network-based approaches aim to learn solution strategies directly from data. One of the earliest neural formulations of TSP was the Hopfield–Tank model (Hopfield & Tank, 1985), which framed the problem as energy minimization in a neural network. Although not data-driven, it marked an early attempt to use neural computation for combinatorial optimization. However, this approach lacked scalability. Recent data-driven methods generally fall into two categories: reinforcement learning (RL), where tours are constructed autoregressively via learned policies (Khalil et al., 2017; Deudon et al., 2018; Kool et al., 2018), and supervised learning (SL), which adopts a two-stage formulation: neural networks generate local preferences, while explicit search procedures build the final solution (Joshi et al., 2019). However, RL methods often suffer from sparse rewards and high training variance, while SL approaches require ground-truth solutions during training, which is computationally expensive due to the NP-hard nature of TSP.

In both RL and SL approaches to the TSP, some form of search is involved, either through learned policies or explicit heuristics. Most RL methods are also autoregressive, generating tours sequentially in a fixed order. Why might we want to move away from this approach? Many combinatorial optimization problems exhibit strong natural structure; in the case of the TSP, this is the shortest Hamiltonian cycle, which inherently constrains the solution space. Recent work (Min et al., 2023) shows that such structure can be exploited through unsupervised, non-autoregressive (NAR) learning, but their framework still requires local search heuristics to assemble final solutions. Building on this, Min & Gomes (2023) formulated TSP as permutation learning with the Gumbel-Sinkhorn operator, though again relying on refinement strategies akin to search. Importantly, Min et al. (2023) demonstrated that unsupervised learning can already reduce the search space, these developments naturally raise the question of whether it is possible to learn high-quality TSP solutions entirely without supervision, search, or autoregressive decoding.

In summary, nearly all data-driven TSP methods, whether supervised, reinforced, or unsupervised, still rely on search. This reliance reveals a core barrier: achieving a learning-based search-free

paradigm for TSP remains a major open challenge, as it would provide direct evidence of neural networks' inherent ability to solve combinatorial problems.

Here, we propose a different perspective on learning combinatorial structures. Rather than treating structure as the output of a post-hoc search, we explore the idea that *structural inductive bias can replace explicit search*. Building on the unsupervised learning for TSP (UTSP) framework, we formulate the TSP as a permutation learning problem: the model directly generates a Hamiltonian cycle using a permutation matrix (Min et al., 2023; Min & Gomes, 2023). Our fully unsupervised, non-autoregressive method requires no optimal training data, search or rollouts. Instead, we train the model using a Gumbel-Sinkhorn relaxation of permutation matrices, followed by hard decoding via the Hungarian algorithm at inference time. This enables solutions to emerge directly from learned structure alone. We further introduce Hamiltonian cycle ensembles that train multiple models on distinct cyclic shift variants and select the best tour across them, thereby reducing long-tail failure.

We demonstrate that our model consistently outperforms classical baselines, including the nearest neighbor algorithm and farthest insertion, across a range of instance sizes. The structural inductive bias encoded in our model alone generates high-quality solutions without explicit search procedures. Our findings suggest that in combinatorial optimization, appropriately designed structural constraints can serve as effective computational mechanisms, offering a complementary paradigm to conventional search approaches. This suggests that structure itself may be sufficient to guide optimization in certain combinatorial problems.

## 2 BACKGROUND: UNSUPERVISED LEARNING FOR THE TSP

The TSP asks to find the shortest route that visits each city exactly once and returns to the starting point. Given $n$ cities with coordinates $x \in \mathbb{R}^{n \times 2}$, we want to find a permutation $\sigma \in S_n$ that minimizes the total tour length:

$$\min_{\sigma \in S_n} \sum_{i=1}^{n} d(x_{\sigma(i)}, x_{\sigma(i+1)}), \tag{1}$$

where $d(\cdot, \cdot)$ is typically the Euclidean distance $d(x_i, x_j) = \|x_i - x_j\|_2$, and we define $\sigma(n+1) := \sigma(1)$ to ensure the tour returns to the starting city.

To enable optimization over Hamiltonian cycles using neural networks, we first introduce a matrix-based representation of permutations. We begin by defining the cyclic shift matrix $\mathbb{V} \in \{0, 1\}^{n \times n}$ for $n \geq 3$ as

$$\mathbb{V}_{i,j} = \begin{cases} 1 & \text{if } j \equiv (i+1) \pmod{n} \\ 0 & \text{otherwise} \end{cases}, \tag{2}$$

for $i, j \in \{0, 1, \ldots, n-1\}$. This matrix has the explicit form:

$$\mathbb{V} = \begin{pmatrix} 0 & 1 & 0 & 0 & \cdots & 0 & 0 \\ 0 & 0 & 1 & 0 & \cdots & 0 & 0 \\ 0 & 0 & 0 & 1 & \cdots & 0 & 0 \\ \vdots & \vdots & \vdots & \vdots & \ddots & \vdots & \vdots \\ 0 & 0 & 0 & 0 & \cdots & 0 & 1 \\ 1 & 0 & 0 & 0 & \cdots & 0 & 0 \end{pmatrix}. \tag{3}$$

The matrix $\mathbb{V}$ represents the canonical Hamiltonian cycle $1 \to 2 \to 3 \to \cdots \to n \to 1$, where each row has exactly one entry equal to unity, indicating the next city in the sequence. More generally, a matrix $\mathcal{H} \in \{0, 1\}^{n \times n}$ represents a Hamiltonian cycle if it is a permutation matrix whose corresponding directed graph forms a single cycle of length $n$.

The key insight is that any Hamiltonian cycle can be generated from the canonical cycle $\mathbb{V}$ through a similarity transformation (Min & Gomes, 2023). Specifically, if $\mathbf{P} \in S_n$ is any permutation matrix, then $\mathbf{P}\mathbb{V}\mathbf{P}^\top$ represents a Hamiltonian cycle obtained by reordering the nodes according to permutation $\mathbf{P}$. Given a distance matrix $\mathbf{D} \in \mathbb{R}^{n \times n}$ where $\mathbf{D}_{ij}$ represents the distance between cities $i$ and $j$, the TSP objective becomes finding the optimal permutation matrix that minimizes:

$$\min_{\mathbf{P} \in S_n} \langle \mathbf{D}, \mathbf{P}\mathbb{V}\mathbf{P}^\top \rangle, \tag{4}$$

where $\langle \mathbf{A}, \mathbf{B} \rangle = \text{tr}(\mathbf{A}^\top \mathbf{B})$ denotes the matrix inner product. The inner product $\langle \mathbf{D}, \mathbf{P} \mathbb{V} \mathbf{P}^\top \rangle$ is the distance of the Hamiltonian cycle represented by $\mathbf{P} \mathbb{V} \mathbf{P}^\top$.

Since finding the optimal discrete permutation matrix is NP-hard and backpropagating through discrete variables is non-differentiable, we relax the problem by replacing the hard permutation matrix $\mathbf{P}$ with a soft permutation matrix $\mathbb{T} \in \mathbb{R}^{n \times n}$. Following (Min et al., 2023; Min & Gomes, 2023), we use a Graph Neural Network (GNN) to construct $\mathbb{T}$ and optimize the loss function:

$$\mathcal{L}_{\text{TSP}} = \langle \mathbf{D}, \mathbb{T} \mathbb{V} \mathbb{T}^\top \rangle. \tag{5}$$

The soft permutation matrix $\mathbb{T}$ approximates a hard permutation matrix. Here, the Hamiltonian cycle constraint is implicitly enforced through the structure $\mathbb{T} \mathbb{V} \mathbb{T}^\top$. This enables gradient-based optimization to find good approximate solutions, while naturally incorporating both the shortest path objective and the Hamiltonian cycle constraint. The GNN learns to generate a soft permutation matrix $\mathbb{T}$ that, when used in the transformation $\mathbb{T} \mathbb{V} \mathbb{T}^\top$, yields a soft adjacency matrix representing a Hamiltonian cycle. This approach provides a non-autoregressive, unsupervised learning (UL) framework without sequential decision-making or ground truth supervision, enabling efficient end-to-end training directly from the combinatorial optimization objective.

## 3 FROM SOFT PERMUTATION $\mathbb{T}$ TO HARD PERMUTATION $\mathbf{P}$

To obtain a hard permutation matrix $\mathbf{P}$ from the GNN output, we follow the method proposed in (Min & Gomes, 2025). We use the Gumbel-Sinkhorn operator (Mena et al., 2018), which provides a differentiable approximation to permutation matrices during training. At inference time, we extract a discrete permutation via the Hungarian algorithm. Following the UTSP model (Min et al., 2023), the GNN processes geometric node features $f_0 \in \mathbb{R}^{n \times 2}$ (city coordinates) along with an adjacency matrix $A \in \mathbb{R}^{n \times n}$ defined by:

$$A = e^{-\mathbf{D}/s}, \tag{6}$$

where $\mathbf{D}$ is the Euclidean distance matrix and $s$ is a scaling parameter. The GNN generates logits $\mathcal{F} \in \mathbb{R}^{n \times n}$ that are passed through a scaled hyperbolic tangent activation:

$$\mathcal{F} = \alpha \tanh(f_{\text{GNN}}(f_0, A)), \tag{7}$$

where $\alpha$ is a scaling parameter, and $f_{\text{GNN}} : \mathbb{R}^{n \times 2} \times \mathbb{R}^{n \times n} \to \mathbb{R}^{n \times n}$ is a GNN that operates on node features $f_0$ and the adjacency matrix $A$.

The scaled logits are passed through the Gumbel-Sinkhorn operator to produce a differentiable approximation of a permutation matrix:

$$\mathbb{T} = \text{GS}\left(\frac{\mathcal{F} + \gamma \epsilon}{\tau}, l\right), \tag{8}$$

where $\epsilon$ is i.i.d. Gumbel noise, $\gamma$ is the noise magnitude, $\tau$ is the temperature parameter which controls relaxation sharpness, and $l$ is the number of Sinkhorn iterations. Lower values of $\tau$ yield sharper, near-permutation matrices. At inference, we derive a hard permutation using the Hungarian algorithm:

$$\mathbf{P} = \text{Hungarian}\left(-\frac{\mathcal{F} + \gamma \epsilon}{\tau}\right). \tag{9}$$

The final Hamiltonian cycle is reconstructed as $\mathbf{P} \mathbb{V} \mathbf{P}^\top$, yielding a discrete tour that solves the TSP instance.

## 4 TRAINING AND INFERENCE

Our training objective minimizes the loss function in Equation 5, incorporating a structural inductive bias: the structure $\mathbb{T} \mathbb{V} \mathbb{T}^\top$ implicitly encodes the Hamiltonian cycle constraint, guiding the model toward effective solutions.

During inference, we decode the hard permutation matrix $\mathbf{P}$ using the Hungarian algorithm as previously described n Equation 9. The final tour is obtained directly through $\mathbf{P} \mathbb{V} \mathbf{P}^\top$, which always generates a valid Hamiltonian cycle by construction. Unlike conventional TSP heuristics requiring

local search, our solutions naturally emerge from the learned permutation matrices. The key advantage of this framework lies in its structural guarantee: regardless of the quality of the learned permutation matrix $\mathbf{P}$, the transformation $\mathbf{P}\mathbb{V}\mathbf{P}^{\top}$ will always yield a feasible TSP solution. The optimization process thus focuses entirely on finding the permutation that minimizes tour length, while the constraint is automatically satisfied.[1]

### 4.1 EXPERIMENTAL SETUP

Our experiments span three distinct problem sizes: 20-node, 50-node, and 100-node TSP instances. For each problem size, we perform hyperparameter sweeps to identify the optimal configurations. Training data consist of uniformly distributed TSP instances generated for each problem size, with 100,000 training instances for 20-node, 500,000 training instances on 50-node instances, and 1,500,000 training instances for 100-node instances. All problem sizes use 1,000 instances each for validation and test.

### 4.2 HYPERPARAMETER CONFIGURATION

We conduct parameter exploration through grid search to evaluate our approach. The 20-node instances are trained across all combinations of temperature $\tau \in \{2.0, 3.0, 4.0, 5.0\}$ and noise scale $\gamma \in \{0.005, 0.01, 0.05, 0.1, 0.2, 0.3\}$, resulting in 24 configurations for each size; the 50-node instances are trained across all combinations of temperature $\tau = 5.0$ and noise scale $\gamma \in \{0.005, 0.01, 0.05, 0.1, 0.2, 0.3\}$, while the 100-node instances use only one temperature value $\tau = 5.0$ with an expanded noise scale $\gamma \in \{0.1, 0.2, 0.3\}$, totaling 3 configurations. We employ $\ell = 60$ Sinkhorn iterations for 20-node instances and $\ell = 80$ for 50- and 100-node instances, with training conducted over 300 epochs for 20-node instances and extended to 600 epochs for 50- and 100-node instances to ensure sufficient convergence. For evaluation, tour distances are computed using hard permutations $\mathbf{P}$ obtained via the Hungarian algorithm as described in Equation 9, in contrast to the soft permutation $\mathbb{T}$ used during training.

### 4.3 NETWORK ARCHITECTURE AND TRAINING DETAILS

Following the UTSP model (Min et al., 2023), we employ Scattering Attention Graph Neural Networks (SAGs) with 128 hidden dimensions and 2 layers for 20-node instances, 256 hidden dimensions and 6 layers for 50-node instances, and 512 hidden dimensions and 8 layers for 100-node instances (Min et al., 2022). For TSP-20, we use SAGs with 6 scattering channels and 2 low-pass channels; for TSP-50 and TSP-100, we use SAGs with 4 scattering channels and 2 low-pass channels. We train the networks using the Adam optimizer (Kingma & Ba, 2014) with weight decay regularization $\lambda = 1 \times 10^{-4}$. Learning rates are set to $1 \times 10^{-3}$ for 20-node instances and $2 \times 10^{-3}$ for 50- and 100-node instances. To ensure training stability, we implement several regularization techniques: (i) learning rate scheduling with a 15-epoch warmup period, (ii) early stopping with patience of 50 epochs, and (iii) adaptive gradient clipping to maintain stable gradients throughout the optimization process.

For each problem size, we select the best performing model based on validation performance across all hyperparameter combinations of $\tau$ and noise scale $\gamma$. The model configuration that achieves the lowest validation distance is then evaluated on the corresponding test set.

## 5 EXPERIMENT

Our training loss with respect to the validation distance is shown in Figure 1. The training curves consistently converge across all problem sizes, with the training loss (blue) steadily decreasing and stabilizing over epochs. Notably, there is a strong correlation between training loss reduction and validation TSP distance improvement (red), indicating effective learning without overfitting. On 20-node instances, the model achieves the best validation distance of 405.60 at epoch 283; on 50-node instances, our model achieves its best distance of 627.21 at epoch 550; on 100-node instances,

---

[1]While we use the Hungarian algorithm to obtain hard permutations from each model's soft output, this step is deterministic and not part of any heuristic or tree-based search procedure. We use the term *search* in the sense of explicit exploration or rollout over solution candidates, as in beam search or reinforcement learning.

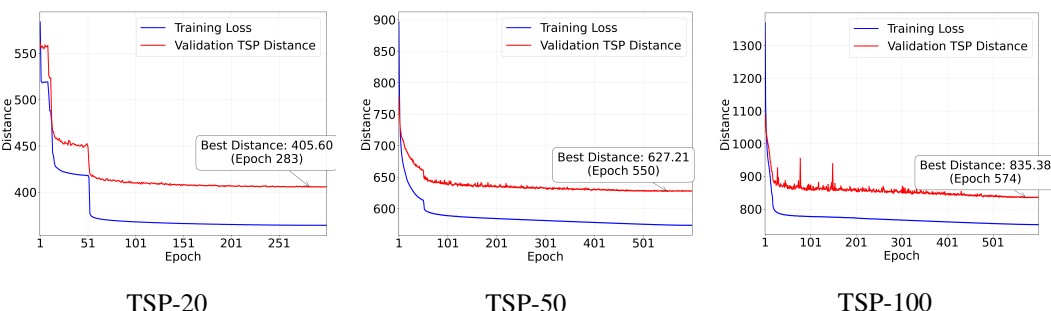

Figure 1: Training history across TSP sizes. Distances are scaled by a factor of 100.

we achieve the best validation distance of 835.38 at epoch 574. Across all scales, the validation performance closely tracks the training loss trajectory, confirming that the model generalizes well and that minimizing the objective in Equation 5 consistently leads to improved TSP solution quality.

## 5.1 LENGTH DISTRIBUTION

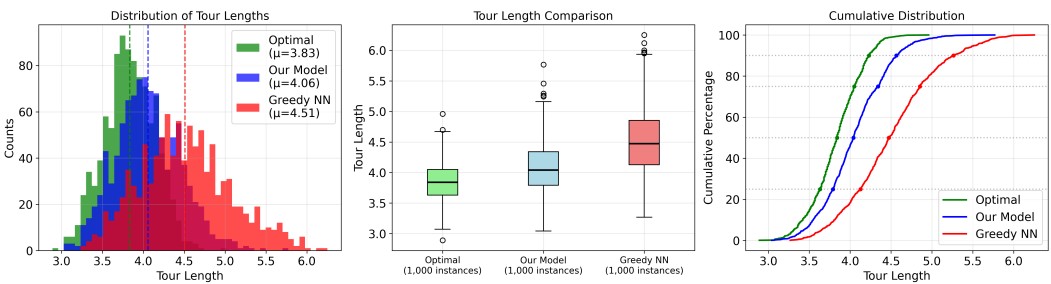

Figure 2: TSP-20 performance comparison showing our model vs. greedy nearest neighbor baseline. Our model achieves 0.45 shorter tour lengths (mean: 4.06 vs. 4.51) with reduced variability and consistently better performance across all percentiles. Results based on 1,000 test instances.

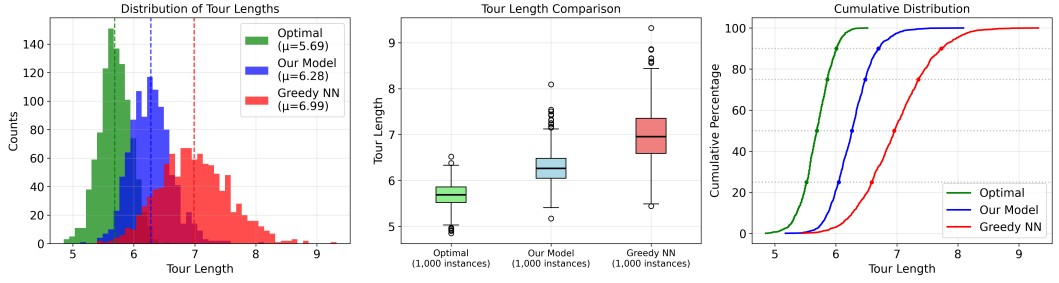

Figure 3: TSP-50 performance comparison showing our model vs. greedy nearest neighbor baseline. Our model achieves 0.71 shorter tour lengths (mean: 6.28 vs. 6.99) with reduced variability and consistently better performance across all percentiles. Results based on 1,000 test instances.

Figures 2, 3, and 4 show the tour length distributions on the test set for 20-, 50-, and 100-node instances, using the model with the lowest validation length across all hyperparameters. Our model consistently outperforms the Greedy Nearest Neighbor (NN) baseline—which constructs tours by iteratively selecting the nearest unvisited node—achieving substantial gains across all problem sizes. The distribution histograms (left panels) reveal that our model produces more concentratedly distributed, shorter tour lengths with mean values of $\mu = 4.06$, $6.28$, and $8.37$ compared to Greedy NN's $\mu = 4.51$, $6.99$, and $9.67$. The box plots (center panels) demonstrate reduced variance and lower median values for our approach, while the cumulative distribution functions (right panels)

show our model achieves better solution quality, with curves consistently shifted toward shorter tour lengths.

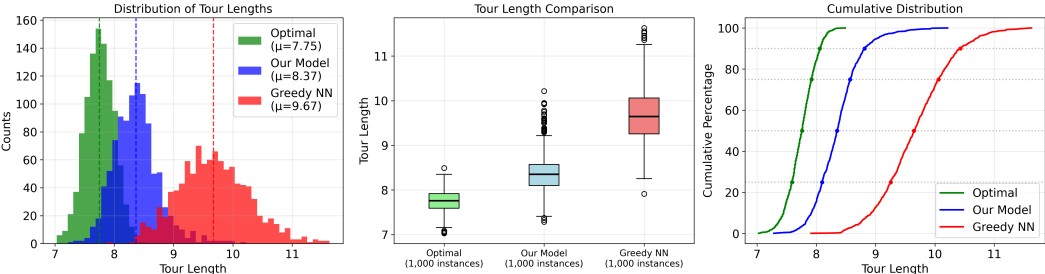

Figure 4: TSP-100 performance comparison showing our model vs. greedy nearest neighbor baseline. Our model achieves 1.30 shorter tour lengths (mean: 8.37 vs. 9.67) with reduced variability and consistently better performance across all percentiles. Results based on 1,000 test instances.

## 5.2 INFERENCE TIME

We evaluate the inference efficiency of our approach by measuring the average time per instance, including the GNN forward pass and construction of the hard permutation as defined in Equation 9. On an NVIDIA H100 GPU with batch size 256, the model achieves inference times of 0.17 ms for TSP-20, 0.15 ms for TSP-50, and 0.40 ms for TSP-100 .

## 5.3 OPTIMALITY GAP

**Optimality Gap Calculation**    The optimality gap is computed as:

$$\text{Gap (\%)} = \left( \frac{\text{Tour Length (Method)} - \text{Tour Length (Optimal)}}{\text{Tour Length (Optimal)}} \right) \times 100, \tag{10}$$

This measures how far a method's tour length deviates from the optimal solution, with smaller gaps indicating better performance.

Table 1: Comparison of tour quality across different heuristics on TSP instances of varying sizes.

| Method | Type | TSP-20 | | TSP-50 | | TSP-100 | |
|---|---|---|---|---|---|---|---|
| | | Tour Len. | Gap | Tour Len. | Gap | Tour Len. | Gap |
| Concorde | Solver | 3.83 | 0.00% | 5.69 | 0.00% | 7.75 | 0.00% |
| Beam search (w=1280) | Search | 4.06 | 6.00% | 6.83 | 20.0% | 9.89 | 27.6% |
| Greedy NN | G | 4.51 | 17.8% | 6.99 | 22.8% | 9.67 | 24.8% |
| Our method | UL, NAR | 4.06 | 6.00% | 6.28 | 10.4% | 8.37 | 8.00% |

Our unsupervised, search-free approach demonstrates competitive performance across TSP instances of varying sizes, achieving optimality gaps of 6.00%, 10.4%, and 8.00% on TSP-20, TSP-50, and TSP-100 respectively (Table 1). Notably, our method matches beam search performance on TSP-20 while significantly outperforming it on larger instances (10.4% vs 20.0% gap on TSP-50, and 8.00% vs 27.6% gap on TSP-100). Our approach also consistently outperforms the Greedy NN baseline across all problem sizes, with the performance advantage becoming more pronounced on larger instances.

These results suggest that our model effectively captures global tour structure and long-range city dependencies, enabling better solutions compared to methods that rely primarily on local, greedy decisions or limited search strategies. The consistent performance indicates that structural inductive biases alone can enable the model to discover competitive combinatorial solutions without supervision or explicit search.

# 6 HAMILTONIAN CYCLE ENSEMBLE

Examining the results in Figures 2, 3, and 4, we observe a long tail distribution where our model yields notably suboptimal solutions on some instances. This suggests that while the model generally performs well, it sometimes generates significantly suboptimal solutions.

To address this limitation, we revisit our training objective in Equation 5: $\mathcal{L}_{\text{TSP}} = \langle \mathbf{D}, \mathbb{T}\mathbb{V}\mathbb{T}^\top \rangle$, where our model learns permutations over the canonical Hamiltonian cycle $\mathbb{V}$. We now propose an ensemble approach utilizing powers of the cyclic shift matrix $\mathbb{V}^k$, where different values of $k$ satisfying $\gcd(k, n) = 1$ generate distinct valid Hamiltonian cycles. Our ensemble strategy trains separate models for each $\mathbb{V}^k$ and selects the minimum tour length solution across all $\varphi(n)$ cycle variants for each test instance, where $\varphi(n)$ is *Euler's totient function*. This leverages diverse Hamiltonian cycle topologies to mitigate long tail behavior, so that when one structure fails, alternatives often succeed, thereby eliminating catastrophic failures.

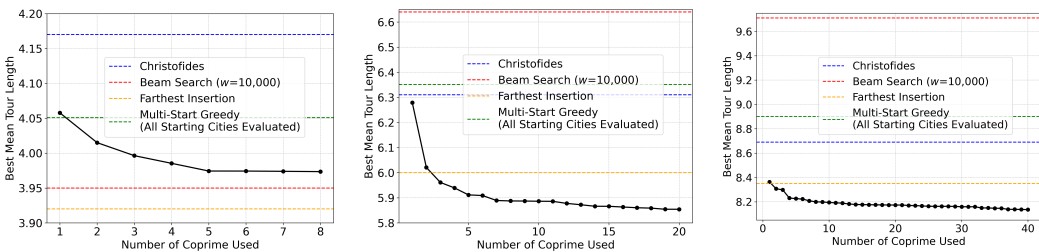

Figure 5: Best mean tour length on TSP instances of different sizes as the number of coprime shifts increases. Using more shift combinations significantly reduces tour length, outperforming the Greedy Multi-Start baseline. Both Christofides and Beam Search bounds are shown for comparison. From left to right: TSP-20, TSP-50, TSP-100.

## 6.1 MAIN THEOREM

**Theorem 6.1** ($\mathbb{V}^k$ Hamiltonian Cycle Characterization). *Let $\mathbb{V}$ be the $n \times n$ cyclic shift matrix and $k \in \mathbb{Z}^+$. Then $\mathbb{V}^k$ represents a Hamiltonian cycle if and only if $\gcd(k, n) = 1$.*

*Proof.* We prove both directions of the equivalence.

**Necessity ($\Rightarrow$):** Suppose $\mathbb{V}^k$ represents a Hamiltonian cycle. Let $d = \gcd(k, n)$. The matrix $\mathbb{V}^k$ corresponds to the mapping $\sigma^k : i \mapsto (i + k) \bmod n$ on the vertex set $\{0, 1, \ldots, n-1\}$.

Consider the orbit of vertex 0 under this mapping:
$$\mathcal{O}_0 = \{0, k \bmod n, 2k \bmod n, \ldots, (m-1)k \bmod n\}, \tag{11}$$
where $m$ is the smallest positive integer such that $mk \equiv 0 \pmod{n}$.

Since $d = \gcd(k, n)$, we can write $k = dk'$ and $n = dn'$ where $\gcd(k', n') = 1$. Then:
$$mk \equiv 0 \pmod{n} \iff dn' \mid mdk' \tag{12}$$
$$\iff n' \mid mk' \tag{13}$$
$$\iff n' \mid m \quad (\text{since } \gcd(k', n') = 1). \tag{14}$$

Therefore, the smallest such $m$ is $m = n' = \frac{n}{d}$, so $|\mathcal{O}_0| = \frac{n}{d}$.

If $\mathbb{V}^k$ represents a Hamiltonian cycle, then all $n$ vertices must lie in a single orbit, which requires $|\mathcal{O}_0| = n$. This implies $\frac{n}{d} = n$, hence $d = 1$, i.e., $\gcd(k, n) = 1$.

**Sufficiency ($\Leftarrow$):** Suppose $\gcd(k, n) = 1$. Then by the argument above, the orbit of vertex 0 has size $\frac{n}{1} = n$. This means the sequence $\{0, k, 2k, \ldots, (n-1)k\}$ modulo $n$ contains all distinct elements of $\{0, 1, \ldots, n-1\}$.

Therefore, $\mathbb{V}^k$ represents a permutation that cycles through all $n$ vertices exactly once, forming a single Hamiltonian cycle. $\qquad\square$

**Corollary 6.2** (Euler's Totient Function Connection). *The number of distinct Hamiltonian cycle matrices of the form $\mathbb{V}^k$ is exactly $\varphi(n)$, where $\varphi$ is Euler's totient function.*

*Proof.* By Theorem 6.1, $\mathbb{V}^k$ represents a Hamiltonian cycle if and only if $\gcd(k, n) = 1$. The number of integers $k \in \{1, 2, \ldots, n\}$ such that $\gcd(k, n) = 1$ is precisely $\varphi(n)$. $\square$

*Remark* 6.3 (Directed vs Undirected Cycles). Note that different values of $k$ may yield distinct *directed* Hamiltonian cycles that correspond to the same *undirected* cycle traversed in opposite directions. For instance, when $n$ is even, $\mathbb{V}^1$ and $\mathbb{V}^{n-1}$ represent the same undirected cycle with opposite orientations. However, each $\mathbb{V}^k$ with $\gcd(k, n) = 1$ defines a unique directed cycle, which is the relevant structure for our ensemble method.

## 6.2 ENSEMBLE TRAINING AND INFERENCE

Our ensemble strategy trains separate models for each valid $\mathbb{V}^k$ matrix. Specifically, we construct $\varphi(n)$ models, each optimizing the modified objective:

$$\mathcal{L}_{\text{TSP}}(k) = \langle \mathbf{D}, \mathbb{T}_{(k)} \mathbb{V}^k \mathbb{T}_{(k)}^\top \rangle, \tag{15}$$

where $\gcd(k, n) = 1$ and $\mathbb{T}_{(k)}$ represents the learned soft permutation matrix corresponding to $\mathbb{V}^k$.

Table 2: Detailed performance comparison of learning-based TSP solvers across different instance sizes (TSP-20/50/100). Metrics include average tour length and optimality gap (%). Results for baseline methods are taken from (Joshi et al., 2019). While all methods use uniformly generated TSP instances, test sets vary slightly across works. Note that many more recent models exist, we select a subset for comparison.

| Method | Type | TSP-20 | | TSP-50 | | TSP-100 | |
| --- | --- | --- | --- | --- | --- | --- | --- |
| | | Tour Len. | Gap | Tour Len. | Gap | Tour Len. | Gap |
| PtrNet (Vinyals et al., 2015) | SL, G | 3.88 | 1.15% | 7.66 | 34.48% | - | - |
| PtrNet (Bello et al., 2016) | RL, G | 3.89 | 1.42% | 5.95 | 4.46% | 8.30 | 6.90% |
| S2V (Khalil et al., 2017) | RL, G | 3.89 | 1.42% | 5.99 | 5.16% | 8.31 | 7.03% |
| GAT (Deudon et al., 2018) | RL, G, 2-OPT | 3.85 | 0.42% | 5.85 | 2.77% | 8.17 | 5.21% |
| GAT (Kool et al., 2018) | RL, G | 3.85 | 0.34% | 5.80 | 1.76% | 8.12 | 4.53% |
| GCN (Joshi et al., 2019) | SL, G | 3.86 | 0.60% | 5.87 | 3.10% | 8.41 | 8.38% |
| POMO (Kwon et al., 2020) | RL,G | 3.83 | 0.12% | 5.73 | 0.64% | 7.84 | 1.07% |
| Concorde | Solver | 3.83 | 0.00% | 5.69 | 0.00% | 7.75 | 0.00% |
| Greedy NN (all start cities) | G | 4.05 | 5.74% | 6.35 | 11.6% | 8.90 | 14.8% |
| Beam search (w=5,000) | Search | 3.98 | 3.92% | 6.71 | 17.9% | 9.77 | 26.1% |
| Beam search (w=10,000) | Search | 3.95 | 3.13% | 6.64 | 17.0% | 9.71 | 25.3% |
| Farthest insertion | Heuristics | 3.92 | 2.35% | 6.00 | 5.45% | 8.35 | 7.74% |
| Christofides | Heuristics | 4.17 | 8.88% | 6.31 | 10.9% | 8.69 | 12.1% |
| **Hamiltonian cycle ensemble** | UL, NAR | 3.97 | 3.52% | 5.85 | 2.81% | 8.14 | 5.03% |

For each $k$, we employ identical training procedures, differing only in the underlying Hamiltonian cycles $\mathbb{V}^k$. We select the model configuration which achieves the lowest validation loss across all hyperparameter combinations for each $k$-specific model. This ensures that each cycle $\mathbb{V}^k$ is properly exploited.

At inference time, we evaluate all $\varphi(n)$ trained models on each test instance. For each model corresponding to Hamiltonian cycles $\mathbb{V}^k$, we decode the hard permutation matrix $\mathbf{P}_{(k)}$ from the learned soft permutation $\mathbb{T}_{(k)}$ using the Hungarian algorithm as previously described in Equation 9. The candidate tour for each ensemble member is obtained directly through $\mathbf{P}_{(k)} \mathbb{V}^k \mathbf{P}_{(k)}^\top$, which always generates a valid Hamiltonian cycle.

For each individual test instance, we then select the solution with minimum tour length across all ensemble members:

$$\text{Tour}_{\text{final}} = \mathbf{P}_{(k^*)} \mathbb{V}^{k^*} \mathbf{P}_{(k^*)}^\top, \tag{16}$$

where $k^* = \arg\min_{k:\gcd(k,n)=1} \langle \mathbf{D}, \mathbf{P}_{(k)} \mathbb{V}^k \mathbf{P}_{(k)}^\top \rangle$ is determined instance-specifically.

This instance-wise selection ensures that each test problem is solved using the most suitable cycle structure from the ensemble, adapting to the particular geometric characteristics of that instance.

Figure 5 demonstrates the effectiveness of this ensemble approach across 20, 50, and 100-node problems respectively. As the number of coprime shifts increases, the mean tour length decreases substantially, with dramatic improvements observed initially that gradually plateau. For TSP-20, using all $\varphi(20) = 8$ coprime shifts reduces mean tour length from 4.06 to 3.97, surpassing both multi-start greedy and beam search performance. Similar trends are observed for TSP-50 and TSP-100, where the ensemble approach achieves mean tour lengths of 5.85 and 8.14 respectively when using all available coprime shifts. This demonstrates that leveraging multiple Hamiltonian cycle structures effectively mitigates the long tail problem while consistently improving solution quality. Here, each permutation learner in our framework is trained with respect to a fixed initial Hamiltonian cycle $\mathbb{V}^k$, which serves as a structural prior guiding solution formation. This initialization anchors the training process, focusing learning on the permutation of the initial Hamiltonian cycle $\mathbb{V}^k$, effectively biasing the model. Since different initial cycles encode distinct structural priors, using an ensemble of models with $\mathbb{V}^k$ promotes diversity and improves overall solution quality.

Despite not using supervision or autoregressive decoding, our method achieves competitive results across all TSP sizes. Our Hamiltonian cycle ensemble approach significantly outperforms classical baselines, achieving optimality gaps of 3.52%, 2.81%, and 5.03% on TSP-20, TSP-50, and TSP-100 respectively, compared to greedy NN (all start cities) search's 5.74%, 11.6%, and 14.8%. We also improve upon beam search variants as the problem size grows, with beam search achieving gaps of 3.13–3.92% on TSP-20, 17.0–17.9% on TSP-50, and 25.3–26.1% on TSP-100.

Among learning-based methods, our approach demonstrates competitive performance. We achieve comparable results to Pointer Networks and S2V. Our method also performs competitively with supervised method (Joshi et al., 2019), achieving 5.03% optimality gap versus 8.38% on TSP-100. Our performance is comparable to the RL-based approaches. Notably, we are competitive with the GAT model of (Deudon et al., 2018) even when it is augmented with 2-OPT local search, a strong post-hoc refinement step. While models such as the attention-based approach by (Kool et al., 2018) leverage RL and autoregressive decoding, our unsupervised, non-autoregressive framework attains similar optimality gaps without requiring either RL training or explicit search procedures.

However, we do not yet match the RL model such as (Kwon et al., 2020), which benefits from exploiting multiple equivalent solutions through parallel rollouts. Our results are also competitive with classical heuristics such as farthest insertion, while offering a fundamentally different approach grounded in structural inductive bias. Overall, our results highlight that non-autoregressive, unsupervised methods can effectively tackle combinatorial optimization problems without sequential decoding. Detailed comparisons are provided in Table 2.

In our experiments, we observe that it is not necessary to employ the full set of $\varphi(n)$ Hamiltonian cycles for effective ensembling. Instead, using a small subset can already yield strong approximations. For example, on TSP-100, selecting the shifts $k \in \{1, 9, 81, 87, 91\}$ achieves an average tour length of 8.18, corresponding to an optimality gap of approximately 5.5%. Furthermore, the inference cost remains practical: whereas a single model requires about 0.40 ms per instance, this five-model subset ensemble takes only 2 ms in total, while still delivering significant improvements in robustness and solution quality.

## 7 CONCLUSION

We present a fully unsupervised, non-autoregressive framework for solving the TSP without relying on explicit search or supervision. By framing the problem as learning permutation matrices that satisfy Hamiltonian cycle constraints via similarity transformations, our approach incorporates structural constraints as inductive biases into the learning process. This formulation enables the model to generate valid tours without sequential decision-making. Our method achieves competitive results and we further demonstrate that ensembles over different Hamiltonian cycles enhance robustness and improve average solution quality, especially on larger problem instances. These results suggest that learned structural biases provide a promising alternative to traditional heuristic search methods by integrating problem structure as an inductive bias in combinatorial optimization.

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

## A  DISCUSSION AND FUTURE WORK

In this paper, we use 24 configurations of $(\tau, \gamma)$ pairs for TSP-20, 6 configurations for TSP-50, and 3 configurations for TSP-100. This limited yet targeted hyperparameter exploration is sufficient to support our central claim: that *structural inductive bias*, when coupled with a permutation-based formulation, can drive the emergence of high-quality solutions in a fully unsupervised, non-autoregressive setting. We restrict our analysis to minimal hyperparameter settings and adopt the same architecture as UTSP (Min et al., 2023), which is sufficiently expressive to illustrate our main claim. While preliminary evidence indicates that performance can be further improved through extensive hyperparameter tuning or architectural variations (e.g., alternative message passing schemes), such enhancements lie outside the scope of our primary contribution and are left for future work.

## B  EFFECTIVENESS OF THE HAMILTONIAN CYCLE ENSEMBLE

Figure 6 shows the tour length distributions produced by models trained with different coprime shifts $\mathbb{V}^k$ for TSP instances of size 20, 50, and 100. Each colored boxplot represents the output distribution from a single model trained on a specific cyclic structure, while the green box on the right shows the ensemble result obtained by selecting the shortest tour across all models for each instance. Notably, while individual models exhibit varying performance and often display long-tail distributions with significant outliers, the ensemble output consistently achieves shorter average tour lengths with reduced variance. This demonstrates that the ensemble strategy effectively mitigates the long-tail failure cases seen in individual models by leveraging structural diversity. Consequently, the ensemble approach leads to more robust and consistent solutions across problem instances.

## C  QUADRATIC UPPER BOUND ON THE OPTIMALITY GAP

**Theorem C.1** (Quadratic upper bound on the optimality gap). *Let $C(\mathbf{P}) := \langle \mathbf{D}, \mathbf{P}\mathbb{V}\mathbf{P}^\top \rangle$ for a cost matrix $\mathbf{D} \in \mathbb{R}^{n \times n}$, a cyclic shift matrix $\mathbb{V} \in \mathbb{R}^{n \times n}$, and a permutation matrix $\mathbf{P} \in \Pi_n$. Let the set of optimal permutations be*

$$\mathcal{O} := \arg \min_{\mathbf{P} \in \Pi_n} C(\mathbf{P}), \qquad C^\star := \min_{\mathbf{P} \in \Pi_n} C(\mathbf{P}). \qquad (17)$$

*Given a (soft) doubly-stochastic matrix $\mathbb{T}$ produced by the model and a hard permutation $\widehat{\mathbf{P}}$ obtained from $\mathbb{T}$ at inference, define*

$$\delta_* := \min_{\mathbf{P} \in \mathcal{O}} \|\mathbb{T} - \mathbf{P}\|_F, \qquad \varepsilon := \|\widehat{\mathbf{P}} - \mathbb{T}\|_F. \qquad (18)$$

*If $\|\mathbb{V}\|_2 \leq 1$ and $\|\mathbb{T}\|_2 \leq 1$, then*

$$C(\widehat{\mathbf{P}}) - C^\star \leq \|\mathbf{D}\|_F \big(2\delta_* + \delta_*^2 + 2\varepsilon + \varepsilon^2\big). \qquad (19)$$

*Proof.* Since $\Pi_n$ is finite, there exists $\mathbf{P}^\dagger \in \mathcal{O}$ such that $\delta_* = \|\mathbb{T} - \mathbf{P}^\dagger\|_F$. We decompose the gap into a "soft" part and a "rounding" part:

$$C(\widehat{\mathbf{P}}) - C^\star = \underbrace{\big(C(\mathbb{T}) - C(\mathbf{P}^\dagger)\big)}_{\text{soft approximation}} + \underbrace{\big(C(\widehat{\mathbf{P}}) - C(\mathbb{T})\big)}_{\text{rounding}}. \qquad (20)$$

**Soft term.** Let $E := \mathbb{T} - \mathbf{P}^\dagger$. Expanding,

$$\mathbb{T}\mathbb{V}\mathbb{T}^\top - \mathbf{P}^\dagger \mathbb{V} \mathbf{P}^{\dagger\top} = E\mathbb{V}\mathbf{P}^{\dagger\top} + \mathbf{P}^\dagger \mathbb{V} E^\top + E\mathbb{V}E^\top. \qquad (21)$$

Using the submultiplicative bounds

$$\|AXB\|_F \leq \|A\|_F \|X\|_2 \|B\|_2, \qquad \|AXB\|_F \leq \|A\|_2 \|X\|_F \|B\|_2, \qquad (22)$$

together with $\|\mathbf{P}^\dagger\|_2 = 1$, $\|\mathbb{V}\|_2 \leq 1$, and $\|E\|_2 \leq \|E\|_F$, we obtain

$$\|E\mathbb{V}\mathbf{P}^{\dagger\top}\|_F \leq \delta_*, \quad \|\mathbf{P}^\dagger \mathbb{V} E^\top\|_F \leq \delta_*, \quad \|E\mathbb{V}E^\top\|_F \leq \delta_*^2. \qquad (23)$$

Thus

$$\|\mathbb{T}\mathbb{V}\mathbb{T}^\top - \mathbf{P}^\dagger \mathbb{V} \mathbf{P}^{\dagger\top}\|_F \leq 2\delta_* + \delta_*^2. \qquad (24)$$

By Cauchy–Schwarz,

$$|C(\mathbb{T}) - C(\mathbf{P}^\dagger)| \leq \|\mathbf{D}\|_F (2\delta_* + \delta_*^2). \qquad (25)$$

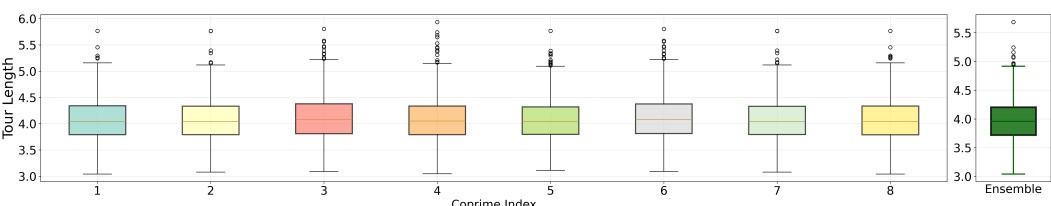

TSP-20: Ensemble vs. 8 individual coprime models

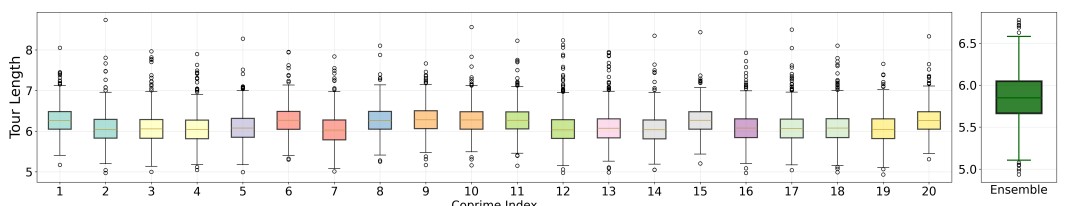

TSP-50: Ensemble vs. 20 individual coprime models

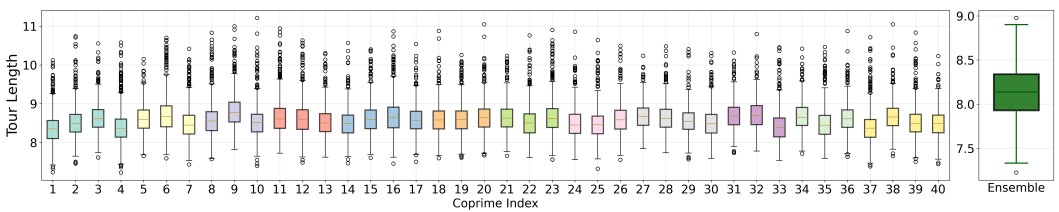

TSP-100: Ensemble vs. 40 individual coprime models

Figure 6: Tour length distributions across individual models and ensemble output for various TSP sizes. Each index corresponds to a model trained using a different coprime shift matrix $\mathbb{V}^k$, where $\gcd(k, n) = 1$. The ensemble result (rightmost box in green) selects the minimum-length tour across all coprime-specific models for each instance.

**Rounding term.** Let $\Delta := \widehat{\mathbf{P}} - \mathbb{T}$, so $\|\Delta\|_F = \varepsilon$. Expanding,

$$\widehat{\mathbf{P}}\mathbb{V}\widehat{\mathbf{P}}^\top - \mathbb{T}\mathbb{V}\mathbb{T}^\top = \Delta\mathbb{V}\mathbb{T}^\top + \mathbb{T}\mathbb{V}\Delta^\top + \Delta\mathbb{V}\Delta^\top. \tag{26}$$

Using the same bounds and $\|\mathbb{T}\|_2 \le 1$, $\|\mathbb{V}\|_2 \le 1$,

$$\|\Delta\mathbb{V}\mathbb{T}^\top\|_F \le \varepsilon, \quad \|\mathbb{T}\mathbb{V}\Delta^\top\|_F \le \varepsilon, \quad \|\Delta\mathbb{V}\Delta^\top\|_F \le \varepsilon^2, \tag{27}$$

hence

$$\|\widehat{\mathbf{P}}\mathbb{V}\widehat{\mathbf{P}}^\top - \mathbb{T}\mathbb{V}\mathbb{T}^\top\|_F \le 2\varepsilon + \varepsilon^2, \tag{28}$$

and by Cauchy–Schwarz,

$$|C(\widehat{\mathbf{P}}) - C(\mathbb{T})| \le \|\mathbf{D}\|_F(2\varepsilon + \varepsilon^2). \tag{29}$$

**Combine.** By the triangle inequality,

$$C(\widehat{\mathbf{P}}) - C^\star \le \|\mathbf{D}\|_F\big(2\delta_* + \delta_*^2 + 2\varepsilon + \varepsilon^2\big). \tag{30}$$

$\square$

**Lemma C.2** (Spectral norm of $\mathbb{V}$). *Let $\mathbb{V} \in \{0, 1\}^{n \times n}$ be the cyclic shift matrix defined in Equation 3. Then*

$$\|\mathbb{V}\|_2 = 1. \tag{31}$$

*Proof.* The matrix $\mathbb{V}$ is a permutation matrix corresponding to a cyclic shift. Permutation matrices are orthogonal, i.e. $\mathbb{V}^\top \mathbb{V} = I$. Hence, all eigenvalues of $V$ have absolute value 1, and

$$\|\mathbb{V}\|_2 = \sqrt{\lambda_{\max}(\mathbb{V}^\top \mathbb{V})} = \sqrt{\lambda_{\max}(I)} = 1. \tag{32}$$

Equivalently, $\mathbb{V}$ is diagonalizable by the discrete Fourier transform, with eigenvalues $\{e^{2\pi i k/n} : k = 0, \ldots, n-1\}$, all lying on the unit circle. Thus the spectral norm of $\mathbb{V}$ is exactly 1. □

**Lemma C.3** (Spectral norm of $\mathbb{T}$). *If $\mathbb{T}$ is doubly-stochastic, then $\|\mathbb{T}\|_2 \leq 1$.*

*Proof.* By the Birkhoff–von Neumann theorem, any doubly-stochastic $\mathbb{T}$ can be written as a convex combination of permutation matrices: $\mathbb{T} = \sum_k \alpha_k \mathbf{P}_k$, with $\alpha_k \geq 0$ and $\sum_k \alpha_k = 1$. The spectral norm is convex, hence

$$\|\mathbb{T}\|_2 = \Big\| \sum_k \alpha_k \mathbf{P}_k \Big\|_2 \leq \sum_k \alpha_k \|\mathbf{P}_k\|_2 = \sum_k \alpha_k \cdot 1 = 1, \tag{33}$$

since each permutation matrix $\mathbf{P}_k$ is orthogonal and thus has spectral norm 1. □

*Remark* C.4 (Interpretation). $\delta_*$ measures how close the learned soft matrix $\mathbb{T}$ is to *some* optimal permutation in $\mathcal{O}$, so the bound handles non-uniqueness naturally. When there are symmetries (e.g. reversed cycles, relabelings), $\delta_*$ will be the distance to the closest such symmetry, which tightens the bound compared to fixing an arbitrary $\mathbf{P}^\dagger$.

# D RANDOMNESS BY HARDWARE PERTURBATION INFERENCE

We introduce Hardware Perturbation Inference (HPI), a simple yet effective technique that leverages the inherent non-determinism of low-level numerical operations to generate diverse inference outcomes without modifying the model or introducing explicit stochasticity. Even when using the same GPU architecture (e.g., NVIDIA H100), small numerical discrepancies can arise from differences in fused multiply–add (FMA) kernel execution and TensorFloat-32 (TF32) rounding modes. These subtle perturbations may propagate through the computation, leading to slightly different outputs. HPI exploits this phenomenon to produce multiple candidate solutions for the same problem instance, which can then be ensembled to improve robustness and solution quality—all without requiring changes to the model parameters or training procedure.

In our experiments, we apply HPI on NVIDIA H100 GPUs by toggling the use of FMA operations under TF32 precision. Specifically, we compare inference with TF32+FMA enabled versus disabled, which yields distinct perturbations in the numerical pathways and consequently different solutions for the same input instance $\mathbb{V}^k$. By combining these outputs in an ensemble, we observe further improvements in solution quality: on the TSP-100 benchmark, the ensemble reduces the optimality gap to 8.10.

While hardware-level perturbations provide a simple mechanism for generating diversity, there are many other ways to introduce randomness to further enhance ensemble performance. We leave a broader discussion of such strategies for future work.

# E ZERO-SHOT GENERALIZATION

We propose a zero-shot evaluation strategy inspired by (Min & Gomes, 2025), leveraging *dummy nodes*. As an example, consider testing on a TSP instance with 95 cities using a model trained on TSP-100. To construct such a test case, we randomly select 5 *parent nodes* from the 95 cities and introduce 5 additional dummy nodes, each placed very close to one of the selected parents. This augmentation produces an effective 100-node instance, which we then solve using the TSP-100 model.

If the resulting tour connects each dummy node directly to its parent node, we merge them to recover a valid tour for the original 95-city problem. If this condition is not met, we repeat the process by re-sampling the parent and dummy nodes.

Using this strategy, our model achieves a mean tour length of $8.24$ across $1,000$ unseen test instances, compared to $9.49$ for the greedy baseline. For reference, the optimal mean tour length is $7.57$. These results demonstrate that our dummy-node construction enables effective zero-shot transfer across problem sizes while maintaining competitive performance.

