# OpenReview forum: "Structure As Search: Unsupervised  Permutation Learning for Combinatorial Optimization"
_ICLR.cc/2026/Conference — ICLR 2026 Conference Withdrawn Submission_

### Official Review · Reviewer_71RX · 2025-10-19

**Soundness:** 2
**Presentation:** 3
**Contribution:** 1
**Rating:** 2
**Confidence:** 4

**Summary:**

This paper focuses on solving the Traveling Salesman Problem (TSP), a classic NP-hard combinatorial optimization problem. It proposes a fully unsupervised, non-autoregressive framework that abandons explicit search; the key innovation lies in formulating TSP as a permutation learning problem, using similarity transformations on Hamiltonian cycles to enable the model to approximate permutation matrices via continuous relaxation, and adopting Gumbel-Sinkhorn relaxation during training and the Hungarian algorithm for hard decoding at inference time. Additionally, a Hamiltonian cycle ensemble strategy is introduced, training multiple models on different cyclic shift variants to mitigate long-tail failures, supported by theoretical results. Experimentally, across 20-node, 50-node, and 100-node TSP instances, the method outperforms classical heuristics like the greedy nearest neighbor algorithm, achieves competitive optimality gaps compared to learning-based methods, and the ensemble strategy further improves solution quality while maintaining efficient inference speed.

However, in general, the proposed method is incremental, and the experiments are not competitive against recent models on solving TSP.

**Strengths:**

- The paper proposes to use unsupervised learning approach to solve TSP, aiming to alleviate the reliance on the labels of supervised learning, as well as the sample-inefficiency of reinforcement learning.

**Weaknesses:**

1. As for the motivation of this paper elaborated in line 40, the accquire of  ground truth labels used for supervised learning for TSP is computationally expensive due to NP hard nature. However, it is not the case as some of heurisitic solvers, e.g. LKH, could be used to obtain solution labels  quite fast with very high quality.
2. The proposed approach is incremental, where the main approach using gumbel-sinkhorn to solve TSP is already used in previous literature, e.g.[1].
3. The baselines are weak with only simple heuristics. The scale in the experiments only covers TSP of nodes of 100, where the main benchmark of recent papers covers 1000 and 10000.

In summary,  the unsupervised learning approach is not convincing enough without innovative approaches or competitive results.

[1] Wang, Runzhong, et al. "Linsatnet: the positive linear satisfiability neural networks." International Conference on Machine Learning. PMLR, 2023.

**Questions:**

See details in the weakness.
The most important concern is that, is it necessary to develop unsupervised learning approach given the cheap and high-quality TSP labels given by heuristics, as well as the much better quality of solutions in supervised solvers in much scalable TSP settings?

---

### Official Review · Reviewer_mMVF · 2025-10-22

**Soundness:** 2
**Presentation:** 2
**Contribution:** 2
**Rating:** 2
**Confidence:** 4

**Summary:**

This paper introduces a refinement of the UTSP, which approaches the Travelling Salesperson Problem using unsupervised learning, by removing the need for a search algorithm to construct a solution which satisfies the constraints of the problem. The paper combines techniques from previous works to learn a soft permutation matrix and extract solutions using the Hungarian algorithm. The paper is well presented, however due to the lack of comparison to other unsupervised baselines and unclear contributions, it is recommended that this paper be rejected unless these concerns can be addressed.

**Strengths:**

S1. The paper is well written and easy to understand, with straightforward and consistent notation.

S2. The method achieves competitive performance with the compared baselines at relatively small problem scales.

**Weaknesses:**

W1. The justification for the paper is that the proposed method is able to remove the requirement for search and autoregression. However, the justification is unclear as it does not make a case for *why* search should be removed, or why autoregression is disadvantageous. It is also unclear how search is necessarily involved in an RL method, as typical model-free RL methods do not perform a lookahead in the search space.

W2. The paper would benefit from a clear explanation of the novel contributions, as it is presently unclear whether the contribution is a new method or a combination of existing techniques.

W3. A major weakness of the paper is the lack of comparison with the UTSP work on which it is based, meaning it is not possible to determine whether any improvement has been made. Furthermore it is missing comparison to any other unsupervised approaches to the TSP. It is concerning that the most recent baseline in Table 2 is from 2020, and there is no justification as to why the more recent models have not been included.

W4. The inference time analysis in Section 5.2 is not useful as it lacks any baseline comparisons.

W5. Some of the analysis in the main part of the paper (e.g. the start of section 5 including Figure 1) do not seem to contribute to the argument of the paper, and would be more suitable in the appendix. Conversely, some assertions are made in the paper that are supported by analysis that can only be found in the appendix, and would be more suitable in the main part of the paper (e.g. Figure 6, supporting the suggestion that "leveraging multiple Hamiltonian cycle structures effectively mitigates the long tail problem").

**Questions:**

C1. The paper is positioned as advantageous in requiring no search-based methods, though it does not explain why search should be avoided. What exactly are the drawbacks of methods which incorporate search that are addressed by using the Gumbel-Sinkhorn operator and the Hungarian algorithm instead?

C2. Table 2 does not provide any comparisons to other UL approaches, e.g. [2, 3], including the one on which this paper is based [1]. Why have these been omitted?

C3. In a pure search-based method, the optimality gap is a result of the search not finding the optimal solution. In a method that uses neural function approximation, the optimality gap is from the approximation. There are a number of baseline methods which use a combination: a neural network outputs a distribution and a search recovers the best solution satisfying the constraints [1, 4, 5]. In this case the optimality gap can be attributed to both parts of the architecture. Since this work builds on [1], can you provide an ablation of your method which uses search instead of the Hungarian algorithm? It would be interesting to characterise the contribution of the Hungarian algorithm to the improvement in the optimality gap. Similarly the inference time is of interest, as the Hungarian algorithm requires $O(n^3)$ time and would likely be slower than approximate search methods.

C4. In Section 6, the authors mention that the method sometimes generates significantly suboptimal solutions. Do other methods, e.g. search-based methods or GNN+search also exhibit this behaviour? I would imagine that there is a chance that the search could entirely fail to recover a valid solution; is the proposed method guaranteed to find a solution?

C5. In Figure 5, how is the set of coprimes chosen for each size? Similarly, in the last paragraph of Section 6.2, how were the values $k \in \{1, 9, 81, 87, 91\}$ chosen?

C6. How does the ensemble method, which chooses the best result of $\varphi(n)$ trained models, compare with training one model for $\varphi(n)$ times more epochs? Or similarly for $m$ models in the ensemble if $m < \varphi(n)$.

C7. Are theorem 6.1 and corollary 6.2 being presented as novel contributions? These are simple properties of cyclic groups.

C8. The studied instances are relatively small compared to other neural approaches to the TSP, which often include instances of up to 1000 cities. Furthermore it appears that the model can only operate on a fixed input size (notwithstanding tricks such as those discussed in Appendix E). Can you point out the part of the architecture that requires a fixed size? Is it only training cost restricting evaluation on larger instances?

Minor comments:

C9. In Figure 6, the plot would be clearer if the plot for the ensemble model used the same scale as those of the coprime models.

### References
[1] Min, Yimeng, Yiwei Bai, and Carla P. Gomes. "Unsupervised learning for solving the travelling salesman problem." _Advances in neural information processing systems_ 36 (2023): 47264-47278.

[2] Gaile, Elīza, et al. "Unsupervised training for neural tsp solver." _International Conference on Learning and Intelligent Optimization_. Cham: Springer International Publishing, 2022.

[3] Sciandra, Lorenzo, et al. "Graph Convolutional Branch and Bound." _arXiv preprint arXiv:2406.03099_ (2024).

[4] Joshi, Chaitanya K., et al. "Learning the travelling salesperson problem requires rethinking generalization." _arXiv preprint arXiv:2006.07054_ (2020).

[5] Ma, Jiale, et al. "COExpander: Adaptive Solution Expansion for Combinatorial Optimization." _Forty-second International Conference on Machine Learning_.

---

### Official Review · Reviewer_MhR8 · 2025-10-23

**Soundness:** 3
**Presentation:** 2
**Contribution:** 3
**Rating:** 4
**Confidence:** 3

**Summary:**

This paper introduces a novel approach to solving the Traveling Salesman Problem (TSP) by leveraging its Hamiltonian Cycle formulation and relaxing the permutation matrix constraint. The authors propose learning a *soft permutation* by parameterizing it with a Graph Neural Network (GNN) and iteratively applying the Gumbel-Sinkhorn operator. During inference, a *hard permutation* is obtained via the Hungarian algorithm. To mitigate local minima in larger instances, the method employs an ensemble of Hamiltonian cycles.

**Strengths:**

- **Novelty and Contribution:** The proposed method is innovative and offers a meaningful contribution to the field of combinatorial optimization.

- **Reproducibility:** The authors provide detailed hyperparameters used during training, which enhances the reproducibility of their approach.

**Weaknesses:**

1. **Clarity and Presentation:**

    - Key concepts, such as the Gumbel-Sinkhorn operator, are not adequately introduced in the main text. At minimum, the authors should refer readers to the appendix or provide a brief explanation.

    - Technical terms like `gcd()` (line 332) and Euler’s totient function are used without explanation, which may confuse readers unfamiliar with these concepts.

    - The table on line 404 lacks a caption, and the best-performing method is not highlighted in bold, making it difficult to interpret results at a glance.

2. **Terminology and Definitions:**

    - The term *“unsupervised learning for CO”* is introduced but never clearly defined. The authors should clarify what “unsupervised” means in this context and justify why reinforcement learning (RL) is excluded from this category. If unsupervised learning is defined as training without solution data, RL would arguably fall under this umbrella.

3. **Visualization:**

    - **Figure 1:** The plots would benefit from the inclusion of baseline methods to provide reference values for what constitutes a “good” distance.

**Questions:**

1. **Model Architecture:** Why was a GNN chosen over alternative architectures, such as Transformers? Given that the adjacency matrix is fully connected, is the GNN formulation computationally efficient?

2. **Computational Resources:** How much memory is required during training? The repeated application of the Gumbel-Sinkhorn operator suggests significant computational overhead—could the authors quantify this?

3. **Inference Efficiency:** What is the runtime of the Hungarian algorithm for obtaining hard permutations? Are there specific early-stopping criteria employed to optimize this step?

4. **Gradient Stability:** How are gradients distributed throughout the method? Backpropagating through Equation 8 appears prone to vanishing or exploding gradients—have the authors observed or addressed this issue?


* * *

**Decision:** I will begin with a **weak reject** but am open to revising my score if the authors adequately address the weaknesses and questions raised above.

---

### Official Review · Reviewer_sMER · 2025-10-30

**Soundness:** 3
**Presentation:** 3
**Contribution:** 3
**Rating:** 6
**Confidence:** 2

**Summary:**

The paper proposes a learning-based search-free method for the traveling salesman problem (TSP). TSP is formulated as a permutation learning problem, and the output of the proposed model is a Hamiltonian cycle using a permutation matrix. The permutation learning problem is further relaxed by replacing the hard permutation matrix with a soft one. Then the loss of the proposed model is the relaxed TSP’s objective function. To obtain a hard permutation matrix as the output, the authors first use the Gumbel–Sinkhorn operator when training, which provides a differentiable approximation to permutation matrices, and then extract a discrete permutation via the Hungarian algorithm when inferencing. To address the limitation that the model sometimes generates suboptimal solutions, they further propose an ensemble approach utilizing the cyclic shift matrix.

**Strengths:**

1. The description of the proposed method is generally clear, though there are a few typos.
2. The solution produced by the proposed method is guaranteed to be feasible.
3. The paper demonstrates that solving TSPs without search is possible, addressing a major open challenge in TSP research.

**Weaknesses:**

1. Some definitions should be clearer. For example, the definition of $S_n$ is missing.
2. Some typos: line 161, “described n Equation 9” -> “described in Equation 9”.
3. The experiment results are preliminary.

**Questions:**

1. In line 138, why $f_{GNN}: R^{n \times 2} \times R^{n \times n} -> R^{n \times n}$? According to standard matrix multiplication, it should be $R^{n\times2} \times R^{2\times n} → R^{n \times n}$ or $R^{n \times n} \times R^{n \times 2} → R^{n \times 2}$.
2. The work appears to be largely based on UTSP. Why don’t the experimental comparisons include this method?
3. Appendix A states that the current experimental results are preliminary and that hyperparameter tuning is insufficient. Why not present final results after thorough hyperparameter tuning? Is the tuning prohibitively expensive?

---

### Note · Authors · 2025-11-19

I have read and agree with the venue's withdrawal policy on behalf of myself and my co-authors.